# Predicting factors that determine patients' satisfaction with post-operative pain management following abdominal surgeries at Komfo Anokye Teaching Hospital, Kumasi, Ghana

**Priscilla Felicia Tano**[1,2]*, **Felix Apiribu**[1], **Emile Kouakou Tano**[2], **Adwoa Bemah Boamah Mensah**[1], **Veronica Millicent Dzomeku**[1], **Isaac Boateng**[3]

**1** Department of Nursing, Faculty of Allied Health Science, College of Health Science, Kwame Nkrumah University of Science and Technology, Kumasi, Ghana, **2** Department of Surgery, Komfo Anokye Teaching Hospital, Kumasi, Ghana, **3** Department of Physiology, University of Cape Coast, Cape Coast, Ghana

* pftano@gmail.com

## Abstract

### Introduction

Poorly controlled postoperative pain has been known to be characterized by longer post-operative care, longer hospital stays with increased readmission rates, and decreased patient satisfaction. Post-operative pain has been continuously addressed in the past three (3) to four (4) decades and has been shown that 20 to 80% of post-operative patients suffer ineffective pain management.

### Objective

The study was aimed at assessing the factors that may predict the satisfaction of patients with early postoperative pain management following abdominal surgeries at the Komfo Anokye Teaching Hospital, Kumasi.

### Methodology

A descriptive cross-sectional study was conducted among patients who had undergone abdominal surgeries between October 2019 and December 2019 at the Komfo Anokye Teaching Hospital. Structured questionnaires based on the IPO-Q were used to obtain responses from the patients. Descriptive and Inferential statistical analysis were employed in analyzing the data obtained from the respondents of the study.

### Results

138 patients were involved in this study. The mean age of patients in the study was 45.81 (±16.81) years. A higher percentage, 58.7% of the patients were males. 39.1% had completed their tertiary level of education. The majority (50.7%) of the patients had had

**Data Availability Statement:** Data are available from figshare repository. https://doi.org/10.6084/m9.figshare.13239449.

**Funding:** The author(s) received no specific funding for this work.

**Competing interests:** The authors have declared that no competing interests exist.

persistent pain for more than three (3) months. The satisfaction of the patients with the post-operative pain management received was generally high among a significant majority of the patients. Meanwhile, among the factors that influence the satisfaction of the patients with the post-operative pain management received, type of analgesia and pain relief methods (Pearson Coefficient = 0.523, p-value <0.05), patient's ability to request more pain relief, (Pearson Coefficient = 0.29, p-value <0.05), patient's access to information about their pain treatment options from the Nurses (Pearson coefficient = -0.22, p<0.05), were the only predictors of satisfaction in patients.

## Conclusion

This study found out that patients were generally satisfied with the post-operative pain management offered by their healthcare providers although the degree of satisfaction depended largely on the type of analgesia and pain relief methods, the ability to request for more pain relief, and access to information on pain treatment.

## Introduction

Management of postoperative pain that is effective has become a major health care concern for practitioners as well as patients who undergo surgical processes [1]. Negative psychological and physiological outcomes for patients can be the downstream effect that may result in transformation into chronic pain, financial burden impacts on health care systems, and relevant decrement in quality of life of the patient when postoperative pain is under-treated [2–4]. The report on the prevalence of acute postoperative pain is known to vary widely and is associated with various adverse consequences [5].

Postoperative pain that is not controlled properly is characterized by longer post-operative care, longer hospital stays with increased readmission rates, and decreased patient satisfaction [6]. After both minor and major surgeries, most patients go through post-operative pain that is acute [7]. According to clinical experiences, the treatment of post-operative pain is not easy to adequately achieve since pain is a complex and subjective experience [8]. Identifying the type of surgery can help in managing effective postoperative pain. Abdominal surgeries are rated among the procedures with the highest pain intensities [9].

Postoperative pain management is an integral part of nursing practice and thus, nurses should have a holistic approach to post-operative care to improve the effectiveness of pain management as well as the satisfaction of patients in this postoperative pain management. Postoperative pain has been continuously addressed in the past three (3) to four (4) decades and has been shown that 20 to 80% of post-operative patients suffer ineffective pain management [5]. A lot of research has been conducted in the developed countries on postoperative pain management [10] whereas, in the developing countries, postoperative pain is understudied especially in Africa. Studies however suggest that the management of postoperative pain remains inadequate [11, 12].

Globally, there has been a force driving the management of postoperative pain, however, postoperative pain is still undertreated especially in the developing countries with the incidence ranging from 47–100% [13]. Worldwide, ineffective treatment of post-operative pain becomes a problem for the patient, the healthcare institution, and the society at large. In Ghana, pain management is an important concern in all health facilities. The treatment of

pain is still inadequate despite the use of pharmacological agents for the management of pain [14]. Pain assessment plays an important role in providing the quality of care for patients after abdominal surgeries. Inadequate assessment and management of postoperative pain can cause sleep disturbances and mobilization difficulties, restlessness, irritability, aggression, and perhaps most importantly, unnecessary levels of distress and suffering to the patient which in turn hinders patients' satisfaction of pain management [15]. Again, ineffective postoperative pain control can lead to chronic pain as well as a decrease in the quality of life [9, 14, 16].

It seems postoperative assessment and management have not had the necessary consideration in developing countries since adequate data for the management of postoperative pain are not accessible but patients who undergo surgical procedures still battle with pain control. It is, therefore, necessary to investigate more into this situation and appreciate the factors influencing the satisfaction of patients' post-operative pain management. Clarification for the unending high levels of post-operative pain as well as suggestive ways of enhancing its management will be detailed.

## Materials and methods

### Study design, site, and population

A descriptive cross-sectional design with a quantitative approach was employed in conducting the study in the surgical unit of the Komfo Anokye Teaching Hospital (KATH). KATH is a tertiary hospital with a 1200- bed capacity that takes direct referrals from 12 out of the 16 administrative regions in Ghana. It also receives patients from neighboring countries like Ivory Coast and Eastern Faso. It is the second-largest hospital in the country. The general surgery ward where participants were recruited has four wards of which females are admitted in two (2) wards with 16-bed capacities each and males being admitted in the other two (2) wards with 18-bed capacities each. The study population comprised patients who were 18 years and above and had undergone general abdominal surgeries within the first 72-hours post-surgery at the surgical wards. The surgeries were performed between October 2019 and December 2019.

### Sample size and sampling technique

A convenience sampling technique was adopted in sampling 138 patients who went through general abdominal surgeries at KATH. The sample size was estimated based on a 95% confidence interval, a 10% estimated proportion and, a 5% error rate.

### Data collecting techniques

Data was collected between October 2019 and December 2019, patients aged 18 years and above who were within 72 hours after abdominal surgery and had signed the consent form were included in the study. However, patients who had had polytrauma experience after undergoing abdominal surgery were excluded. A written letter was obtained from the head of the department of surgery where it was sent to various wards where the study was to be done. Permission was given by the in-charge of the ward to commence the collection of data. Data were collected at the patient's bedside after the ward in-charge clears the patient to be stable. Research assistants were coached on how to recruit participants for the study and on assessing patients' ability to understand the information given about the study and to partake in the study. Patients were made aware of the main aim of the study as well as its nature, risks, and processes. Written consents were gotten from participants after an extensive explanation of the study before data collection was done. Participation in the study was purely voluntary and participants were given the option of opting out of the research at any point. Patients were

assured that opting out of the study would not affect the care rendered to them in the hospital. Participants were also assured of the confidentiality of data gathered.

## Ethical consideration

Certificate of registration (RD/CR18/231) was obtained from the Research Unit of the Komfo Anokye Teaching Hospital. Ethical clearance and consent forms were sought from the Committee on Human Research Publication and Ethics, KNUST, with the clearance number CHPRE/AP/061/19 before the commencement of data collection.

## Instrument

The International Pain Outcome questionnaire (IPO-Q) which was developed by Pain Out [17] based on the American Pain Society questionnaire [18] was used as the instrument for data collection. It ensures the provision of a tool for assessing POP and improving health care quality. The IPO-Q includes major features of measuring outcomes in acute pain. They include the severity of pain, pain interference with mood, physical functions, side effects of treating pain, care perception as well as satisfaction of treating pain after post-surgery. Also, are questions using non-medicine methods of pain relief, and the presence and intensity of preoperative chronic pain. The IPO-Q typically employs 11- point (0–10) numerical rating scale items and binary items. Two items focusing on time in severe pain and pain relief received use percentage scales [18]. Studies that have used IPO Questionnaire have shown satisfactory Cronbach's alpha of 0.85–0.88 [17]. A p-value of less than 0.05 was statistically significant. Socio-demographic variables of patients were also collected. Patients' medical records were reviewed to collect data on the type of surgery, type of anesthesia, duration of surgery, the type of analgesics administered, and dosage.

## Data handling

After data collection, all questionnaires were stored in files with access restricted to the principal investigator only. This aided in guaranteeing the confidentiality of the participants. Questionnaires were given codes for archiving and easy retrieval. Data from the questionnaires were double-checked before entry was done into a computer by the principal investigator. Data entered on the computer were protected with a password.

## Data analysis

Analysis of data was conducted with IBM SPSS statistics 25. The sample was described using descriptive analysis, where means and standard deviations (SDs) were used for continuous variables and frequencies and percentages for categorical variables. A mean interference score on a numerical rating scale (NRS 1–10) was calculated, with higher scores reflecting greater pain intensity and greater interference with physical activities. To establish the correlation strength, and direction between the variables, a correlation analysis was used by estimating Pearson's correlation ratios. The relationships among demographic variables and procedural characteristics of patients as predictors of satisfaction with post-operative pain management were examined by Pearson's correlation, Chi-square test, and one-sample t-tests. All tests were conducted at a level of significance of $P < 0.05$.

## Results

The mean age of the participants was 45.81 (±16.81) years with the majority of the patients belonging to middle adults aged 36–55 years. More than half of the patients, 58.7% were males

with 41.3% being females. A higher percentage, 31.9% of the participants who had formal education had completed the tertiary level, with only 7.2% who had no form of formal education in their lifetime. Moreover, the majority of the patients were Christians 79.0% with only 21.0% being Muslims. The study also recorded the type of surgery the patients received. A higher proportion of the total sample population, 44.9% had exploratory laparotomy performed on them, followed by 30.4% who received herniorrhaphy, 19.6% who received appendicectomy, 4.3% who received cholecystectomy, and only 0.7% who received gastrectomy. After the surgery, 97.1% responded to have received general anesthesia, with only 4 (2.9%) who received spinal anesthesia. For most of the patients, 98.6% received a combination therapy of opioids and NSAIDS as their post-operative analgesia with only 1.4% who received only opioids postoperative analgesia. Regarding the severity of persistent pain assessed with the numerical rating scale (NRS), 73.1% had Moderate (NRS 4–7) persistent pain, 23.9% had Severe persistent pain (NRS 8–10), whereas 3% had Mild (NRS 0–3) persistent pain. Regarding the location of persistent pain, the majority of patients 52.2% had experienced the pain both at the site of the surgery and elsewhere, and 44.8% experienced the pain only at the site of the surgery. The rest of the patients (3%) however felt the persistent pain elsewhere away from the site of the surgery (Table 1).

The pain characteristics of the patients were recorded on a numerical rating scale (NRM 0–10). The patients further described how the pain interferes with their physical activities with a mean scale of 6.89 (±2.48) of interference with activities in bed, a mean scale of 5.93 (±3.58) of interference with breathing deeply or coughing, and a mean scale of 4.74 (±2.27) of interference with sleep. The mean scale of the interference of pain with activities out of bed was also 6.02 (±3.26). The study further recorded how pain affects the emotion of the patients and causes them to be anxious and helpless. The mean scale of the respondents on how pain causes them to be anxious was 3.53 (±2.55) with a mean scale of 3.41 (±2.57) for the feeling of helplessness. The pain was also described to have had side effects on the patients. The patients described nausea as an effect of the pain with a mean scale of 1.57 (±2.106), drowsiness with a mean scale of 2.69 (±2.04), itchiness with a mean scale of 0.25 (±1.23) and, dizziness with a mean scale of 2.46 (±1.96). On an NRS of 0–10, the mean scale of patient's response to their involvement in the decision concerning the pain treatment options was 0.03 (±0.24) with a mean scale of 6.97 (±0.94) for their satisfaction with the pain treatment they received (Table 2). A substantial majority of 74.64% of the participants responded to have had moderate satisfaction with the entire procedures involved in the postoperative pain management they received followed by 22.46% who were extremely satisfied with the post-operative pain management received. Only 2.90% however were not satisfied with the entire process involved in postoperative pain management (Fig 1).

Based on the demographic and procedural characteristics as predictors of satisfaction in patients with the post-operative pain management, there was no statistically significant difference of satisfaction across the various levels of education (p>0.05), though patients who had pursued education up to the secondary level were found to be extremely satisfied with the POP management. Moreover, Christians were found to be extremely satisfied with pain management as compared to Muslims. This difference however was statistically insignificant (p>0.05). Also, there was no statistically significant difference of satisfaction across the type of surgery the patients received though most of the patients who received Laparotomy were found to be extremely satisfied. There was no statistically significant difference in satisfaction between males and females, though extreme satisfaction was found more in males than females (p>0.05). No significant association was found between age range and satisfaction with pain treatment (p>0.05). Extreme satisfaction however was found more in patients aged 36–55 years. A statistically significant association was found between the prescribed post-operative

**Table 1. Demographics and procedural characteristics of participants (n = 138).**

| Variable | Frequency (%) |
|---|---|
| *Gender* | |
| **Male** | 81 (58.7%) |
| **Female** | 57 (41.3%) |
| *Age Range* | |
| **Young adults (18–35)** | 41 (29.7%) |
| **Middle adults (36–55)** | 62 (44.9%) |
| **Older adults (>55)** | 35 (25.4%) |
| *Formal educational level* | |
| **None** | 10 (7.2%) |
| **Primary** | 41 (29.7%) |
| **Secondary** | 44 (31.9%) |
| **Tertiary** | 43 (21.2%) |
| *Religion* | |
| **Christian** | 109 (79%) |
| **Muslim** | 29 (21%) |
| *Type of Surgery* | |
| **Appendicectomy** | 27 (19.6%) |
| **Cholecystectomy** | 6 (4.3%) |
| **Exploratory laparotomy** | 62 (44.9%) |
| **Gastrectomy** | 1 (0.7%) |
| **Herniorrhaphy** | 42 (30.4%) |
| *Type of Anaesthesia* | |
| **General** | 134 (97.1%) |
| **Spinal** | 4 (2.9%) |
| *Postoperative analgesia* | |
| **Opioids only** | 2 (1.4%) |
| **NSAIDs only** | 0 (0%) |
| **Combination therapy** | 136 (98.6%) |
| *Persistent pain ≥3 months* | |
| **Yes** | 68 (49.3%) |
| **No** | 70 (50.7%) |
| *The severity of persistent pain (0–10)* | |
| **Mild (0–3)** | 2 (3.0%) |
| Moderate (4–7) | 49 (73.1%) |
| **Severe (8–10)** | 16 (23.9%) |
| *Location of persistent pain* | |
| **Site of surgery** | 62 (44.8%) |
| **Elsewhere** | 4 (3.0%) |
| **Both** | 72 (52.2%) |

n: number of cases.

analgesia and satisfaction with postoperative pain treatment (p<0.05). Extreme satisfaction was found to be significantly higher in patients who received IV Morphine & IM Diclofenac (Table 3).

Regarding the outcome of pain management as a predictor of satisfaction in patients with postoperative pain treatment, there was a statistically significant positive correlation (Pearson

**Table 2. Postoperative pain characteristics and management (n = 138).**

| Domain | Variable (scale) | Mean (SD) | Median (range) |
|---|---|---|---|
| *Pain interference with physical activities* | Activities in bed (0–10) | 6.89 (±2.48) | 7.48 |
| | Breathing deeply or coughing (0–10) | 5.93 (±3.58) | 7.40 |
| | Sleeping (0–10) | 4.74 (±2.27) | 5.21 |
| | Activities out of bed (0–10) | 6.02 (±3.26) | 7.24 |
| *Pain effect on emotions* | Anxiety (0–10) | 3.53 (±2.55) | 4.16 |
| | Helplessness (0–10) | 3.41 (±2.57) | 4.03 |
| *Side effects* | Nausea (0–10) | 1.57 (±2.106) | 0.63 |
| | Drowsiness (0–10) | 2.69 (±2.04) | 3.20 |
| | Itching (0–10) | 0.25 (±1.23) | 0.05 |
| | Dizziness (0–10) | 2.46 (±1.96) | 2.78 |
| *Decisions on Treatment* | Participation in pain treatment decisions (0–10) | 0.03 (±0.24) | 0.03 |

SD: standard deviation n: number of cases.

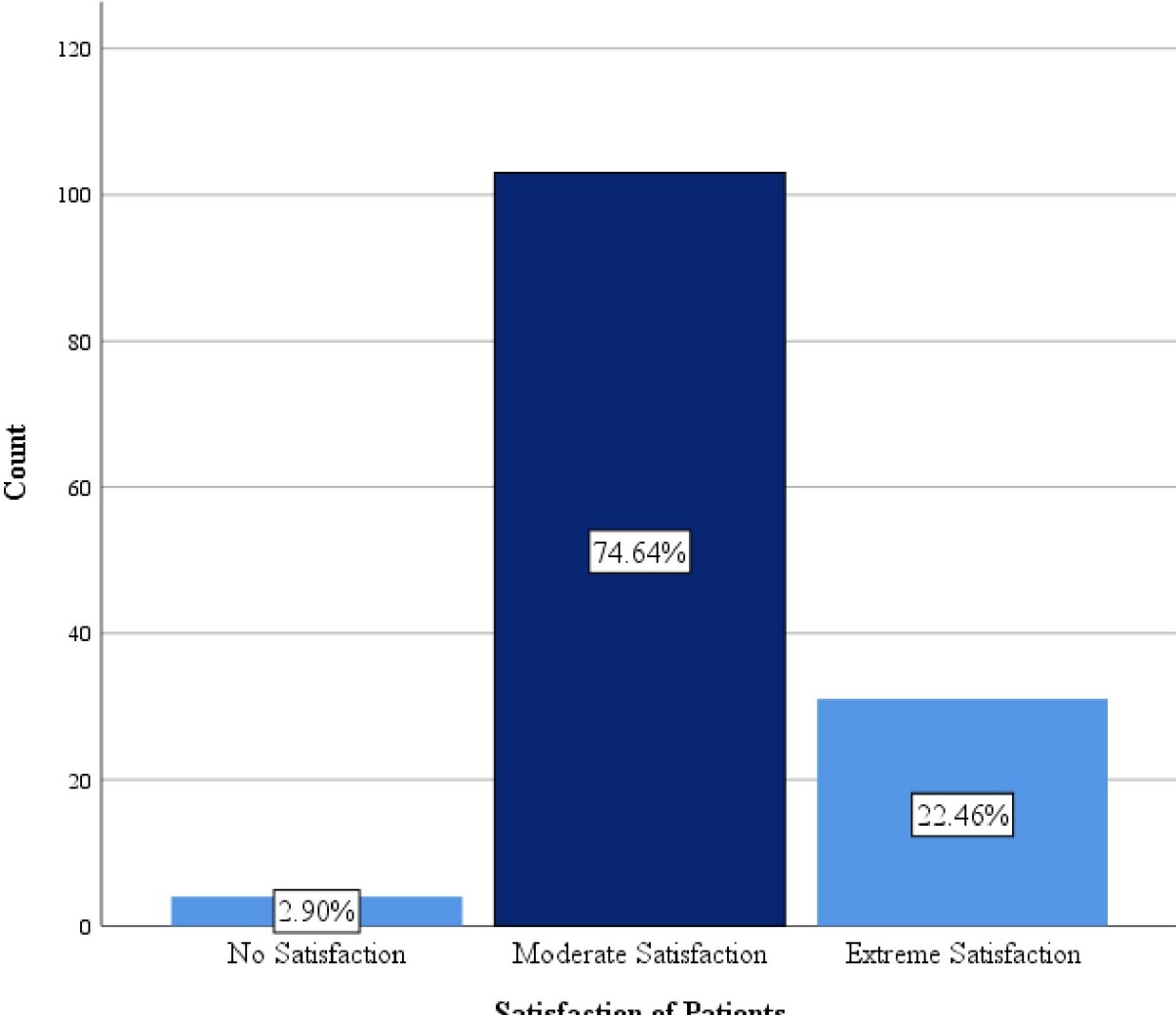

**Fig 1. Patient satisfaction with pain treatment received since surgery.**

**Table 3. Demographic and procedural characteristics of patients as predictors of satisfaction of post-operative pain management in patients.**

| Modality | | The satisfaction of patients with POP Management (n) | | | |
| --- | --- | --- | --- | --- | --- |
| | | No Satisfaction | Moderate Satisfaction | Extreme Satisfaction | p-value |
| *Level of Education* | Primary | 0 | 33 | 8 | 0.95 |
| | Secondary | 3 | 28 | 13 | |
| | Tertiary | 1 | 36 | 6 | |
| | None | 0 | 6 | 4 | |
| *Religion* | Christian | 3 | 80 | 26 | 0.75 |
| | Muslim | 1 | 23 | 5 | |
| *Type of Surgery* | Hernia repair | 1 | 32 | 9 | 0.83 |
| | Exploratory laparotomy | 3 | 47 | 12 | |
| | Appendicectomy | 0 | 17 | 10 | |
| | Cholecystectomy | 0 | 6 | 0 | |
| | Gastrectomy | 0 | 1 | 0 | |
| *Type of Anaesthesia* | General | 4 | 99 | 31 | 0.51 |
| | Spinal | 0 | 4 | 0 | |
| *Gender* | Male | 4 | 57 | 20 | 0.16 |
| | Female | 0 | 46 | 11 | |
| *Age Range* | 18–35 | 2 | 27 | 12 | 0.17 |
| | 36–55 | 2 | 47 | 13 | |
| | 55 and above | 0 | 29 | 6 | |
| *Prescribed post-operative Analgesia* | IM Pethidine | 0 | 0 | 1 | 0.00 |
| | IM Pethidine & IM Diclofenac | 0 | 1 | 0 | |
| | IM Morphine | 0 | 8 | 1 | |
| | IV Morphine & IM Diclofenac | 0 | 16 | 15 | |
| | IV Morphine & IV Paracetamol | 2 | 75 | 11 | |
| | IV Morphine & IV Diclofenac | 0 | 0 | 1 | |
| | IV Morphine & Suppository Diclofenac | 0 | 1 | 0 | |
| | IM Morphine & IM Diclofenac | 2 | 2 | 2 | |

n: number of cases, POP: Postoperative Pain, p<0.05 significant.

Coefficient = 0.523, p-value <0.05) between the Pain relief received and Satisfaction. Also, a statistically significant positive correlation (Pearson Coefficient = 0.29, p-value <0.05) was identified between the patient's ability to request more pain relief and Satisfaction. Surprisingly, the patient's access to information about their pain treatment options from the Nurses produced a statistically significant negative correlation with their satisfaction in the whole process of postoperative pain management (Pearson coefficient = -0.22, p<0.05). Meanwhile, there was no statistically significant association of satisfaction with any side effects like Nausea, drowsiness, itching, and dizziness (p>0.05). (Table 4).

The study further accessed nursing-related pain care if they could be predictors of satisfaction in patients. How often nurses respond to the pain needs of patients, quality of care as judged by the patients, and time of administration of medications had no statistically significant correlation with satisfaction in patients (Table 5).

## Discussion

In patients who undergo surgical procedures in the hospital, post-operative management remains a challenge worldwide. In this present study, 44.9% representing a higher percentage of the sample population had exploratory laparotomy as compared to the others who had

**Table 4. Outcome of pain management as predictors of satisfaction in patients.**

| Satisfaction of Patients | | |
|---|---|---|
| Modality | Correlation Coefficient | p-value |
| Interference of pain with morbidity | 0.03 | 0.76 |
| Anxiety | -0.12 | 0.17 |
| Helplessness | -0.14 | 0.10 |
| Side effect–NAUSEA | -0.17 | 0.05 |
| Side effects–DROWSINESS | -0.10 | 0.22 |
| Side effects–ITCHING | 0.09 | 0.29 |
| Side effects–DIZZINESS | 0.05 | 0.60 |
| Intensity of Pain | -0.07 | 0.40 |
| Pain relief received | 0.52** | 0.00 |
| Request for more pain relief | 0.29** | 0.00 |
| Did you receive any information about your pain treatment options from the nurses? | -0.22* | 0.01 |
| Were you allowed to participate in decisions about your pain treatment as much as you wanted to? | -0.05 | 0.55 |

p<0.05 significant.

herniorrhaphy, appendicectomy, cholecystectomy, and gastrectomy respectively. All the surgeries were classified under major surgeries with no minor surgery recorded. A high percentage of exploratory laparotomy could be due to acute abdomen which most of the patients in our study presented with. This finding disagrees with a study at the University of Gondar Hospital where 84% of the patients had major surgery and the rest of the 16% had minor surgeries (26). The difference could be as a result of the type of population involved in our study in which patients wait for symptoms to exacerbate before coming to the hospital.

In this study, general anesthesia was the most administered type of Anaesthesia in 97.1% of the patients, with 98.6% who received a combination therapy of opioids and NSAIDS as their post-operative analgesia. This study further reveals that a substantial majority of 73.1% of the patients had moderate persistent pain after treatment with analgesia though only 23.9% had severe persistent pain. This could be because the majority of them felt the pain both at the site of the surgery and other different locations away from the site of the surgery. Lemay *et al.*, (2018) in a qualitative study of post-operative pain reported persistent pain in patients following total joint replacement surgery [19]. Furthermore, Strike *et al.*, (2019) in a randomized control trial further reported persistent pain in a transapical aortic valve replacement surgery, though the persistent pain lasted for a week [20].

Based on the interference of the persistent pain with physical activities recorded in a numerical rating scale (NRM 0–10), this present study found out that patients mostly

**Table 5. Nursing-related pain care as predictors of satisfaction in patients.**

| Satisfaction of Patients | | |
|---|---|---|
| Modality | Correlation Coefficient | p-value |
| How often did a nurse respond to your pain needs? | 0.01 | 0.90 |
| How will you rate the quality of care administered by the nurse? | 0.14 | 0.10 |
| Were your medications administered on time | -0.16 | 0.05 |

p<0.05 significant.

complained of sleeping pattern with a mean scale of 4.74 (±2.27), breathing or coughing with a mean scale of 5.93 (±3.58), and other activities out of bed with a mean scale of 6.89 (±2.48), as being interfered by the pain they felt.

Despite the persistent pain of patients in this present study, there were no side effects such as anxiety 3.53 (±2.55), nausea 1.57 (±2.106), drowsiness 2.69 (±2.04), itchiness 0.25 (±1.23), and dizziness 2.46 (±1.96) based on the numerical rating scale (NRM 0–10). This finding could be due to proper management of the pain by the healthcare providers as was also reported by Ireland and Lalkhen (2019) [21].

A higher percentage of the patients in this present study responded to have not been involved in most of the decisions regarding their pain treatment with a mean numerical rating scale of 0.03 (±0.24). This reason could go a long way to affect their satisfaction with the whole process of post-operative pain management. Active participation of patients in their pain care according to McDonall et al., (2016) is important for ensuring safe and high-quality healthcare and ensuring satisfaction with postoperative pain management [22]. The least participation of patients in pain treatment decisions was also reported by Ramia et al., (2017) [23].

The low level of involvement of participants in the treatment decisions affected the overall outcome of the post-operative procedures in this current study as was reported by Ramia and his colleagues. Geeraerts and his colleagues further agree with the findings of this study and reported that patients do not have more chances to take part in managing their pain [13]. In times when patients were involved, the inclusion seemed more concentrated on reporting pain other than treating the pain. Moreover, Lauren and her colleagues agree that the least involve- ment of patients in treatment decisions has implications on the quality of pain management they receive. Engaging patients in their pain management during the period of hospitalization ensures comfort and reduces the potential for complications. This adequately prepares the patients to manage their pain after discharge from the hospital.

The majority of the participants (74.64%) were found to be moderately satisfied with the post-operative pain management they received, though extreme satisfaction was found in only 22.46% of the patients. Stanimir et al., (2016) disagree in a case report, suggesting that there has been a marked decline in desirable psychological and clinical outcomes of post-operative management affecting the quality of life of patients and thus their satisfaction with the whole process [24]. This difference in satisfaction with post-operative pain management could be attributable to the reassurance the patients received before the surgery began.

Drewett, (2008) however agreed in her report that effective management of post-operative pain results in improved patient outcomes and increased patient satisfaction as was found in this present study [25]. This finding in this present study further, disagrees with reports by Gebremedhn et al., (2015) in which patient satisfaction with Anaesthesia services was low [26]. This discrepancy could be due to the patients in their study who were discharged in the first 24 h after surgery, and 10 (5%) who were unconscious postoperatively.

Regarding the factors that may determine satisfaction of patients with post-operative pain management, this study found out that only the type of prescribed analgesia (p<0.05), the type of pain relief (Pearson Coefficient = 0.523, p-value <0.05), the patient's ability to request for more pain relief (Pearson Coefficient = 0.29, p-value <0.05), and access to information about their pain treatment options from the Nurses (Pearson coefficient = -0.22, p<0.05) served as predictors of satisfaction of the patients with the overall process of the post-operative pain management.

The findings of this present study disagree with findings by Farooq et al., (2016) [27] who reported that only the quality of care provided by the acute pain management service team provided positive inputs regarding satisfaction in patients. This difference could be due to the involvement of some of the participants in our study in the process of pain management.

However, Gebremedhn *et al.*, (2015) [26] found consistent results in which 90.4% out of 200 patients who were operated upon were satisfied based on the anaesthesia and type of analgesia received. Furthermore, patients from a study by Hallway *et al.*, (2019) [28] reported minimal to no opioid use after the administration of an opioid-sparing pathway and were still found to be extremely satisfied with the post-operative pain management.

## Conclusion

The findings of this study revealed that despite the persistent pain experienced by the patients, there were no side effects like drowsiness, nausea, dizziness, and itching. The majority of the participants were found to be satisfied with the post-operative management they received. Moreover, the type of prescribed analgesia, the type of pain relief, the patient's ability to request for more pain relief, and access to information about their pain treatment options from the Nurses served as predictors of satisfaction of the patients with the overall process of the post-operative pain management. Based on the findings of this study, nurses and other healthcare providers should be educated maximally on the management and assessment of postoperative pain. Further studies are needed to replicate findings on other surgical cases, and in other health institutions with an emphasis on the implementation of effective postoperative pain management.

## Limitations of the study

This study was mainly a cross-sectional design. This study was only focused on only patients who had undergone abdominal surgeries. Hence, the findings from this study may not apply to other cohorts of participants who had undergone other types of surgery. Again, this study was a single-site study and hence the extent to which the findings are generalized to patients who have undergone abdominal surgeries should be done cautiously.

## Acknowledgments

We would like to express our profound gratitude to all post-operative patients who participated in the study.

## Author Contributions

**Conceptualization:** Priscilla Felicia Tano, Emile Kouakou Tano.

**Data curation:** Priscilla Felicia Tano.

**Formal analysis:** Priscilla Felicia Tano, Isaac Boateng.

**Investigation:** Priscilla Felicia Tano.

**Methodology:** Priscilla Felicia Tano.

**Supervision:** Felix Apiribu, Emile Kouakou Tano.

**Writing – original draft:** Priscilla Felicia Tano.

**Writing – review & editing:** Priscilla Felicia Tano, Adwoa Bemah Boamah Mensah, Veronica Millicent Dzomeku.

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
