## [Decision Letter · Decision Letter 0]

11 Nov 2020

PONE-D-20-32363

Predicting factors that determine patients’ satisfaction with post-operative pain management following abdominal surgeries at Komfo Anokye Teaching Hospital, Kumasi, Ghana.

PLOS ONE

Dear Tano,

Thank you for submitting your manuscript to PLOS ONE. After careful consideration, we feel that it has merit but does not fully meet PLOS ONE’s publication criteria as it currently stands. Therefore, we invite you to submit a revised version of the manuscript that addresses the points raised during the review process.

We look forward to receiving your revised manuscript.

Kind regards,

Emre Bozkurt

Academic Editor

PLOS ONE

Additional Editor Comments:

Congratulations for your well written manuscript. You need to make some revisions for your manuscript before it can be published. The reviewer suggestions were as follows;

Reviewer 1;

There are two major revisions I would like to emphasize.First, the first paragraph of Material and Methods is mainly disconnected and there is no integrity of meaning. To understand the method of this study, it is necessary to read the manuscript three or four times. Material method section should be revised.

Secondly, the current results of the study were not mentioned in the discussion section. The literature was well searched, but no correlation was made with the results found in the study or the differences of the study from the literature were not mentioned in the section

Reviewer2;

The period when the patients were operated on is not mentioned in the article. The period when patients are operated on should be specified in Abstract (Methodology) and Main Text ( Materials and Methods) sections.

The manuscript will be re-evaluated after these revisions.

Journal Requirements:

2. Please include additional information regarding the validation of the modified International Pain Outcome (IPO-Q) questionnaire used in the study and ensure that you have provided sufficient details that others could replicate the analyses. Furthermore, if questionnaire is not under a copyright more restrictive than CC-BY, please include a copy, in both the original language and English, as Supporting Information.

Furthermore in the Methods section, please provide additional details regarding how demographic information was collected.

Reviewers' comments:

Reviewer's Responses to Questions

**Comments to the Author**

1. Is the manuscript technically sound, and do the data support the conclusions?

Reviewer #1: Partly

Reviewer #2: Partly

2. Has the statistical analysis been performed appropriately and rigorously? 

Reviewer #1: I Don't Know

Reviewer #2: Yes

3. Have the authors made all data underlying the findings in their manuscript fully available?

Reviewer #1: Yes

Reviewer #2: Yes

4. Is the manuscript presented in an intelligible fashion and written in standard English?

Reviewer #1: No

Reviewer #2: Yes

5. Review Comments to the Author

Reviewer #1: There are two major revisions I would like to emphasize.

First, the first paragraph of Material and Methods is mainly disconnected and there is no integrity of meaning. To understand the method of this study, it is necessary to read the manuscript three or four times. Material method section should be revised.

Secondly, the current results of the study were not mentioned in the discussion section. The literature was well searched, but no correlation was made with the results found in the study or the differences of the study from the literature were not mentioned in the section.

Reviewer #2: The period when the patients were operated on is not mentioned in the article. The period when patients are operated on should be specified in Abstract (Methodology) and Main Text ( Materials and Methods) sections.

6. PLOS authors have the option to publish the peer review history of their article (what does this mean?). If published, this will include your full peer review and any attached files.

Reviewer #1: No

Reviewer #2: **Yes: **Uygar Demir

---

## [Author Response · Author response to Decision Letter 0]

24 Dec 2020

Department of Nursing

College of Health Sciences

Kwame Nkrumah University of Science and Technology

Kumasi-Ghana

24th December, 2020

Editor-in-Chief

PLOS ONE 

Dear Sir/Madam, 

RESPONSE TO REVIEWERS’ AND EDITORIAL COMMENTS: PONE-D-20-32363

We express our appreciation to the editors and reviewers for the comments and suggestions made in reference to the manuscript that we submitted titled: “Predicting factors that determine patients’ satisfaction with post-operative pain management following abdominal surgeries at Komfo Anokye Teaching Hospital, Kumasi, Ghana”. Authors have revised the manuscript using the reviewers’ and the editorial comments. 

Accordingly, we have provided the response to the comment in the table of changes below, and where the addition has been effected.

Thank you

PRISCILLA FELICIA TANO

CORRESPONDING AUTHOR

Reviewer # Page Number Reviewers’ Comments Authors’ Responses

REVIEWER 1 6-9

19-22 The first paragraph of materials and methods is mainly disconnected and there is no integrity of meaning. To understand the method of this study, it is necessary to read the manuscript three to four times. Materials and methods section should be revised.

The current results of the study were not mentioned in the discussion section. 

The literature was well searched, but no correlation was made with the results found in the study or the differences of the study from the literature were not mentioned in the section.

 We thank the reviewer for this important comment to revise the Materials and methods section. This comment has been addressed. Materials and methods section has been revised from page 6-9, line 122-199

We are grateful to the reviewer for this significant comment to mention the current result of the study in the discussion section. This comment has been addressed in the discussion section in page 19-22, line 288-364.

We appreciate this important comment to correlate or bring out the difference with the result found in the study and the literature. This comment has been addressed in the discussion section in page 19-22, line 288-364.

Reviewer 2; 3,6 The period when the patients were operated on is not mentioned in the article.

The period when patients are operated on should be specified in Abstract (Methodology) and Main Text (Materials and Methods) sections

 We thank the reviewer for this valuable feedback. The authors have included the periods when patients were operated on in the Abstract (Methodology) in page 3 line 57 and also in Materials and methods section in page 6, line 132.

---

## [Editor Report · Decision Letter 1]

19 Jan 2021

PONE-D-20-32363R1

Predicting factors that determine patients’ satisfaction with post-operative pain management following abdominal surgeries at Komfo Anokye Teaching Hospital, Kumasi, Ghana.

PLOS ONE

Dear Dr. Tano,

Thank you for submitting your manuscript to PLOS ONE. After careful consideration, we feel that it has merit but does not fully meet PLOS ONE’s publication criteria as it currently stands. Therefore, we invite you to submit a revised version of the manuscript that addresses the points raised during the review process.

We look forward to receiving your revised manuscript.

Kind regards,

Emre Bozkurt

Academic Editor

PLOS ONE

Additional Editor Comments (if provided):

Congratulations for your well written manuscript. You need to make some revisions for your manuscript before it can be published. The reviewer suggestions were as follows;

Reviewer 1;

There are two major revisions I would like to emphasize.First, the first paragraph of Material and Methods is mainly disconnected and there is no integrity of meaning. To understand the method of this study, it is necessary to read the manuscript three or four times. Material method section should be revised.

Secondly, the current results of the study were not mentioned in the discussion section. The literature was well searched, but no correlation was made with the results found in the study or the differences of the study from the literature were not mentioned in the section

Reviewer2;

The period when the patients were operated on is not mentioned in the article. The period when patients are operated on should be specified in Abstract (Methodology) and Main Text ( Materials and Methods) sections.

The manuscript will be re-evaluated after these revisions.

---

## [Author Response · Author response to Decision Letter 1]

29 Mar 2021

Reviewer # Page Number Reviewers’ Comments Authors’ Responses

REVIEWER 1 The first paragraph of materials and methods is mainly disconnected and there is no integrity of meaning. To understand the method of this study, it is necessary to read the manuscript three to four times. Materials and methods section should be revised.

The current results of the study were not mentioned in the discussion section. 

The literature was well searched, but no correlation was made with the results found in the study or the differences of the study from the literature were not mentioned in the section.

 We thank the reviewer for this important comment to revise the Materials and methods section. This comment has been addressed. Materials and methods section have been revised from page 6-9, line 125-197

We are grateful to the reviewer for this significant comment to mention the current result of the study in the discussion section. This comment has been addressed in the discussion section in page 18-22, line 284-363.

We appreciate this important comment to correlate or bring out the difference with the result found in the study and the literature. This comment has been addressed in the discussion section in page 18-22, line 294-295, 301-302, 324-316, 339-340, 345-346, 356-357.

Reviewer 2; The period when the patients were operated on is not mentioned in the article.

The period when patients are operated on should be specified in Abstract (Methodology) and Main Text (Materials and Methods) sections

 We thank the reviewer for this valuable feedback. The authors have included the periods when patients were operated on in the Abstract (Methodology) in page 3 line 58 and also in Materials and methods section in page 6, line 137.

---

## [Editor Report · Decision Letter 2]

7 May 2021

Predicting factors that determine patients’ satisfaction with post-operative pain management following abdominal surgeries at Komfo Anokye Teaching Hospital, Kumasi, Ghana.

PONE-D-20-32363R2

Dear Dr. Priscilla Felicia Tano

We’re pleased to inform you that your manuscript has been judged scientifically suitable for publication and will be formally accepted for publication once it meets all outstanding technical requirements.

Kind regards,

Ehab Farag, MD FRCA FASA

Academic Editor

PLOS ONE
---

## [Editor Report · Acceptance letter]

14 May 2021

PONE-D-20-32363R2 

Predicting factors that determine patients’ satisfaction with post-operative pain management following abdominal surgeries at Komfo Anokye Teaching Hospital, Kumasi, Ghana. 

Dear Dr. Tano:

I'm pleased to inform you that your manuscript has been deemed suitable for publication in PLOS ONE. Congratulations! Your manuscript is now with our production department. 

Kind regards, 

on behalf of

Dr. Ehab Farag 

Academic Editor

PLOS ONE